# Relationship between Nutrient Intake and Hearing Loss According to the Income Level of Working-Aged Adults: A Korean National Health and Nutrition Survey

**DOI:** 10.3390/nu14081655

**Published:** 2022-04-15

**Authors:** Juhyung Lee, Ji-Hyeon Lee, Chulyoung Yoon, Chanbeom Kwak, Jae-Joon Ahn, Tae-Hoon Kong, Young-Joon Seo

**Affiliations:** 1Department of Biostatistics, Yonsei University, Wonju College of Medicine, Wonju 26426, Korea; dlwngud1995@naver.com (J.L.); fezro@yonsei.ac.kr (C.Y.); 2Department of Otorhinolaryngology, Yonsei University, Wonju College of Medicine, Wonju 26426, Korea; leegh1310@naver.com; 3Research Institute of Hearing Enhancement, Yonsei University, Wonju College of Medicine, Wonju 26426, Korea; 4Laboratory of Hearing and Technology, Research Institute of Audiology and Speech Pathology, College of Natural Sciences, Hallym University, Chuncheon 24252, Korea; cksqja654@gmail.com; 5Division of Speech Pathology and Audiology, College of Natural Sciences, Hallym University, Chuncheon 24252, Korea; 6Division of Data Science, Yonsei University, Wonju 26493, Korea; ahn2615@yonsei.ac.kr

**Keywords:** hearing loss, income level, nutrient intake, working-age adults

## Abstract

The relationship between hearing impairment and nutrition has been extensively investigated; however, few studies have focused on this topic in working-age adults by income level. Herein, we aimed to determine the differences in hearing impairment among working-age adults by income level and identify the nutritional factors that affect hearing loss in various socioeconomic groups. Seven-hundred-and-twenty participants had hearing impairment, while 10,130 had normal hearing. After adjustment for propensity score matching, income and smoking status were identified as significant variables. By assessing the relationship between hearing impairment and nutrient intake by income level using multiple regression analyses, significant nutrients differed for each income category. Carbohydrate and vitamin C levels were significant in the low-income group; protein, fat, and vitamin B1 levels were significant in the middle-income group; and carbohydrates were significant in the high-income group. Income was significantly associated with hearing impairment in working-age adults. The proportion of individuals with hearing impairment increased as income decreased. The association between hearing impairment and nutritional intake also differed by income level. Our findings may enable the establishment of health policies for preventing hearing impairment in working-age adults by income level.

## 1. Introduction

Hearing loss (HL) is a major public health concern affecting an estimated 360–554 million adults and children worldwide [1]. There is a disproportionately high prevalence of HL in low-resource areas, particularly in South Asia. The causes of HL are multifactorial, including low education/socioeconomic status, high noise exposure, race/ethnicity, active smoking and passive smoke, cardiovascular health, diabetes, genetics, and ototoxic drugs [2,3]. Hearing impairment was recently reported to be extensively associated with diet and nutrition in older individuals. As nutrition is essential for determining an organism’s maintenance, growth, reproduction, health, and disease status, proper nutrition was hypothesized as the first step in the prevention and potential repair of hearing damage before an irreversible state is reached. The World Health Organization (WHO) listed prenatal iodine deficiency as a nutritional cause of HL [4]. According to Jung’s systematic review of the association between nutritional factors and HL in Korea, the incidence of HL increased with the lack of even a single micronutrient, such as vitamins A, B, C, D, and E, zinc, magnesium, selenium, iron, and iodine. Higher carbohydrate, fat, and cholesterol intake or lower protein intake has also been associated with poorer hearing status [5]. Suitable guidelines for maintaining a proper nutritional status may prevent some of the causes and burdens of HL. Socioeconomic inequalities in food and nutrient intake have been widely reported in Korea [6]. In particular, income may influence dietary quality owing to food accessibility and availability [7]. Thus, nutritional imbalances may be more prevalent in the low-income group than that in the high-income group [8].

Since the economic status of Korea has markedly changed since 1960, with a transition to Westernized nutritional behavior, a study of the association between income and dietary intake is needed to reveal the factors affecting HL. Thus, this study aimed to investigate the association between income and HL and identify the key nutritional factors that affect HL using cross-sectional data from the fifth Korean National Health and Nutrition Examination Survey (KNHANES). Public health professionals and policymakers should devote effort to increasing food availability, accessibility, and affordability among low-income adults with HL.

## 2. Materials and Methods

### 2.1. Hearing Measurements

The subjects underwent the pure-tone audiometry (PTA) as a standard protocol (ANSI 3.6 type 1 pure tone audiometer, 1989; IEC 645-1 type 1 pure tone audiometer, 1992) in a double-walled single-room audio booth.

Pure-tone thresholds were obtained for air conduction at 250, 500, 1000, 2000, 4000, and 8000 Hz. Audiological data were reported in accordance with the methods recommended by the Hearing Committee of the American Academy of Otolaryngology Head and Neck Surgery, and the average PTA was calculated using the following formula: (5000 Hz + 1000 Hz + 2000 Hz + 4000 Hz)/4. Hearing impairment was defined as an average PTA score of >40 dB.

### 2.2. Collection of Study Data and Participants

Raw data from the fifth (2010–2012) KNHANES were analyzed in this study. The KNHANES consists of health, nutrition, and checkup surveys, and is performed to produce statistics with representation and reliability of national and provincial units of health, health consciousness, behavior, food, and nutrition [9]. Common survey areas for individuals older than 1 year include morbidity, impairment, activity restriction, medical use, and education. Safety awareness, such as fastening seatbelts when driving or wearing helmets when riding a bicycle, and for adults (19 years and older) and adolescents (12–18 years old), economic activity, smoking, drinking, and mental health were also investigated. The quality of life of adults was assessed, and household surveys were conducted for one adult per household based on household type, home ownership, housing behavior, average monthly income, and marital status. The nutritional survey consisted of eating behaviors, dietary supplements, nutritional knowledge, food safety, 24-h food recall, and food intake frequency. Health and check-up surveys were conducted at the mobile check-up center, and a nutrition survey was conducted via in-person visits to the target households. All education, economic activity, morbidity, medical use, and nutrition components of the health survey were evaluated. Among the health survey items, health behaviors such as smoking and drinking were self-reported. Screening surveys were conducted using direct measurements, observations, and sample analysis.

A circular survey of 192 survey units and 3840 households was conducted annually with a total of 25,534 samples, including 8958 in 2010, 8518 in 2011, and 8058 in 2012. A total of 10,850 people were selected as the final participants; thus, 3259 people younger than 20 years or older than 65 years and 11,425 who did not respond to the survey or did not take the test were excluded. The study was approved by Wonju Severance Christian Hospital (IRB number: CR321349).

### 2.3. Income Level

Four income-related variables are included in the data. In several studies, the monthly household income quartile is often chosen to represent income. However, in this study, individual income variables were selected rather than household income as each household member had different personal characteristics based on their occupation or lifestyle. To identify the stark differences between income levels, we select the quintile individual income variable and classified it into very low, low, middle, high, and very high. The low and very low categories were recombined into low, high and very high were recombined into high.

### 2.4. Demographic Characteristics

The individuals were divided into age categories of 20–24, 25–29, 30–34, 35–39, 40–44, 45–49, 50–54, 55–59, and 60–64 years. (Subjects were analyzed in the working-age group, which did not include those over 65 years of age.). The Organization for Economic Cooperation and Development classifies the working-age population as individuals aged 15–64 years. Individuals over 65 years of age were also excluded from the analysis because geriatric HL may act as a confounding factor. When the participants were asked “Have you ever worked in a noisy place such as that with a mechanical sound or generator for more than three months?” their responses were recorded as “yes” or “no”. However, if their response was “no memory”, it was excluded. The participants were asked, “Have you ever been married?” If yes, which of the following is your current marital status: “Spouse present and live together” or “Spouse present but do not live together”? If participants were never married, they were classified as “unmarried.” Furthermore, if their answer was “No spouse due to spouse death” or “No spouse due to divorce”, the Divorced & Bereavement category was selected. If their response was “no memory”, it was excluded. A total of four categories were established for the question “How often do you drink?”: “not at all in the last year”, “less than once a month”, and “once a month” as non-drinkers or “two to four times a week” as drinkers. Smoking status was divided into three categories: “never smoked”, “less than five packs (100 cigarettes)”, and “more than 5 packs (100 cigarettes)”.

### 2.5. Metabolic Syndrome

The KNHANES includes screening items that measure waist circumference, blood pressure, high-density lipoprotein cholesterol (HDL-C), fasting glucose, and triglyceride level, which define metabolic syndrome. Metabolic syndrome was defined according to the diagnostic criteria of the National Cholesterol Education Program Adult Treatment Panel. The diagnostic criteria were abdominal obesity (waist circumference >90 cm for men and >80 cm for women), high-neutrality fat (>150 mg/dL), low HDL-C (<40 mg/dL for men and <50 mg/dL for women), and high blood pressure (130/85 mm Hg or higher).

### 2.6. Nutritional Intake

Nutritional surveys included food safety, food intake frequency, and food intake surveys (24-h recall method). The food intake frequency survey included variables of weekly intake frequency and average intake per serving, and the daily nutrient intake calculated therein is the generated variable. The 2012–2019 food safety survey based on the US Household Food Security/Hunger Survey module. An integrated analysis could not be performed because of these changes. The food intake frequency survey was conducted using a simple food intake frequency survey table consisting of 63 food items until the 2nd year of the 5-year period (2011). However, the validity of the 112 food items from the 3rd year of the 5-year period (2012) was verified. Using a quantitative food intake frequency survey table, the intake frequency and serving amount for each item were investigated. The semi-quantitative food intake frequency survey of 112 food items (2012) and the food intake frequency survey (2010–2011) consisting of 63 food items comprised the items in the two survey tables, although the items were the same. As the responses and frequencies were different, an integrated analysis could not be performed. Therefore, the food intake survey was based on food intake content (24-h recall method) one day before the survey. The variables included energy, water, protein, fat, carbohydrate, fiber, ash, calcium, phosphorus, iron, sodium, potassium, vitamin A, carotene, retinol, vitamin B1, riboflavin, niacin, and vitamin C; these variables represent the sum of the daily nutrient intakes from all foods and foods consumed by an individual during the day.

### 2.7. Data Analysis

The analysis was conducted using SAS 9.4 (SAS Institute Inc., Cary, NC, USA). The *t*-test and chi-square tests were used to determine whether demographic characteristics (including gender, age, come level, noise exposure, marital status, drinking, and smoking) affect HL and to compare prevalence rates. The level was considered significant when the *p*-value was less than 0.05.

To reduce the influence of age, sex, and noise exposure, which are known as risk factors for heating impressions, 1:3 propensity score matching was performed using the three variables to properly determine the effect of other variables on HL.

Simple and multiple regression models were used to determine the association between hearing impairment and demographics, habits, medical histories, and nutritional intakes.

## 3. Results

The participants’ general characteristics according to hearing impairment status are shown in Table 1. As age increased in the hearing impairment group, the proportion of individuals increased, with the highest proportion (39.58%) in the 60–64 years group, revealing a statistically significant difference (*p* < 0.0001). In the no hearing impairment and hearing impairment groups, the proportion of women was significantly higher than that of men (*p* < 0.0001). Noise exposure was also statistically significant (*p* < 0.0001). In the no-hearing impairment group, the proportion of people with a high income was 40.72%. In the hearing impairment group, people with a low income were the most common at 42.36%, but income was not statistically significant (*p* = 0.0508). The number of participants with “unmarried” status was the highest in each group, with statistical significance (*p* < 0.0001). The percentage of non-smokers in the no hearing impairment and hearing impairment groups was 62.38% and 53.19%, respectively.

The proportion of participants who smoked “over 5 packs” of cigarettes in the hearing impairment group was 45.14%, significantly higher than that of the no hearing impairment group (*p* < 0.0001). Alcohol consumption was 56.79% and 46.53% in their groups, respectively, with “2–4 times/month” the most frequent response. In the hearing impairment group, 9.58% drank “≥4 times/week,” which was significantly higher than that in the non-hearing impairment group (*p* < 0.0001). There were more targets without metabolic syndrome that were statistically significant (*p* < 0.0001). Among the items that aided in the diagnosis of metabolic syndrome, waist circumference (*p* = 0.0015), fasting glucose level (*p* < 0.0001), hypertension (*p* < 0.0001), and triglyceride level (*p* < 0.0001) were statistically significant, while HDL-C level was not statistically significant (*p* = 0.3606).

The general characteristics of the study participants on the multiple regression analysis are shown in Table 2. Age, sex, noise exposure, and income level were the statistically significant variables. As age increased, the likelihood of not having HL decreased (OR = 0.642, *p* < 0.0001). In terms of sex, HL was observed more frequently in women than in men (OR = 1.307, *p* = 0.0308). We were able to control for sex confounding factors by implementing PSM. When participants were exposed to noise, their likelihood of not having HL was reduced (OR = 0.779, *p* = 0.0208). Furthermore, the higher the income level, the higher the likelihood of no HL (OR = 1.097, *p* = 0.0395).

The results of the 1:3 PSM using age, sex, and noise exposure, which are known risk factors for hearing impressions, are shown in Table 3. Because the standardized difference of all variables is reduced or maintained after PSM than, before PSM, appropriate matching was achieved.

The general characteristics of the participants according to hearing impairment status in the matched data are shown in Table 4. As age increased in the hearing impairment group, the proportion of participants increased, with the highest proportion (39.58%) in the 60–64 years group, which was statistically significant (*p* < 0.0001). In the no-hearing impairment group, the proportion of people with a high income was 43.8%. In the hearing impairment group, the proportion of people with a low income was 42.36%, and income was a statistically significant variable (*p* = 0.0012). In the no hearing impairment and hearing impairment groups, most participants were “unmarried”; however, the differences relative to the other groups were not statistically significant (*p* = 0.2692).

Among the no hearing impairment group, 56.25% were non-smokers, which was higher than that of the hearing impairment group of 53.19%, respectively. Furthermore, 45.14% of smokers in the hearing impairment group reported “over 5 packs,” a significantly higher proportion than that of the no hearing impairment group (*p* = 0.0444).

The proportion of alcohol consumption between the no hearing impairment and hearing impairment groups was not significant (*p* = 0.4389). There were more targets without metabolic syndrome that were not statistically significant (*p* = 0.6212). Among the factors used to diagnose metabolic syndrome, waist circumference (*p* = 0.6274), HDL-C (*p* = 0.8609), fasting glucose (*p* = 0.5709), hypertension (*p* = 0.5095), and triglycerides (*p* = 0.8656) were non-statistically significant variables.

The general characteristics of the participants in the matched data based on multiple regression analysis are shown in Table 5. Income level and smoking status were not statistically significant variables. Furthermore, the higher the income level, the greater the likelihood of no HL (OR = 0.855, *p* = 0.0013). An increased prevalence of smoking also increased the likelihood of a no HL status (OR = 1.152, *p* = 0.0439).

The differences in nutritional intake between normal hearing and hearing loss groups according to income using simple regression analysis are shown in Table 6. Calcium and vitamin C were significant nutrients in the low-income group. In fact, the mean calcium intake was 506.86 ± 316.97 mg in the no hearing impairment group and 503.76 ± 364.78 mg in the hearing impairment group. The average intake of vitamin C was 112.31 ± 88.02 mg in the no hearing impairment group and 103.42 ± 97.2 mg in the hearing impairment group. Riboflavin was a significant nutrient in the middle-income group. The average intake of riboflavin was 1.29 ± 0.69 mg in the no hearing impairment group and 1.19 ± 0.57 mg in the hearing impairment group. Carbohydrate was a significant nutrient in the high-income group. The mean carbohydrate intake was 334 ± 126.63 g in the no hearing impairment group and 360.86 ± 129.74 g in the hearing impairment group.

The differences in nutritional intake between the normal hearing and HL groups according to income using multiple regression are shown in Table 7. To properly identify the odds ratio, the continuous variable nutrient intake was newly established and analyzed with a nominal variable divided by quartiles. The carbohydrate and vitamin C levels were statistically significant in the low-income group. A lower carbohydrate intake reduced the risk of HL (OR = 0.78, *p* = 0.031), as did a higher vitamin C intake (OR = 1.302, *p* = 0.0074). In the middle-income group, the protein, fat, and vitamin B1 levels were statistically significant. A lower protein intake reduced the risk of HL (OR = 0.638, *p* = 0.0442), as did a higher fat intake (OR = 1.332, *p* = 0.0432). A reduced vitamin B1 intake decreased the risk of HL (OR = 0.746, *p* = 0.0401). In groups with a high income, a lower carbohydrate intake decreased the risk of HL (OR = 0.783, *p* = 0.0397).

The distribution of HL degree by income group is shown in Figure 1. More than half of the participants had “moderate” HL. To compare the degree of HL according to income level, the imbalance in HL degree must be resolved. Therefore, weight was employed as the reciprocal of the ratio of HL degree and statistically significant using the chi-squared test (*p* < 0.0001). “Profound” was the most common in “Low-income,” “Severe” in “Middle-income,” and “Moderate” in “High-income”.

## 4. Discussion

This study confirmed the correlation between income and HL and verified the major nutrients affecting HL from the 5th (2010–2012) KNHANES. Based on the reported epidemiological and clinical factors, income was the most significant risk factor for HL. Carbohydrate and vitamin C intakes were significant in the low-income group. Lower carbohydrate and higher vitamin C intake reduced the risk of HL (OR = 0.78, *p* = 0.031; and OR = 1.302, *p* = 0.0074, respectively). In the middle-income group, the protein, fat, and vitamin B1 intakes were statistically significant variables. This is similar to a Korean study that reported that vitamin C intake was associated with HL and a French study that found that vitamin B1 intake reduced the risk of HL [9,10]. In an Austrian study, carbohydrate intake increased the incidence of HL [11], a finding that is also consistent with the results of this study. Nutritional diets play an important role in reducing the effects of HL. The role of nutrition in preventing HL indicates that the increased consumption of antioxidant vitamins can reduce HL. Antioxidants, such as vitamin A and C, reduced risk of HL [12]. Protein intake is reportedly associated with hearing discomfort in the older Korean population [13]. Another study reported the effects of vitamin E on HL [14,15,16,17]. In this study, since vitamin E intake could not be confirmed, its association with HL could not be determined, but it is thought that a further analysis would be possible with the intake of food groups containing nutrients such as nuts. Carbohydrate intake was statistically significant, even in the high-income group, highlighting the relationship between hearing and nutrition. However, in the low-income group, vitamin C is considered a deficient nutritional component.

In several studies, overall dietary quality was better in the high-income group than in the low-income group. A previous study reported that the intakes of protein, calcium, phosphorus, potassium, and vitamin C were lower in the low-income than in the high-income group. In women, the protein and niacin intakes were lower in the low-income group. Men in the lowest income group ate fewer dairy products, while women in this group ate fewer fruits, fish, or shellfish. These findings indicate that income is related to nutrient intake [18,19]. Nutritional deficiencies may be detrimental to HL patients. Thus, a low income negatively affects HL [20,21]. A study in Brazil found that children from high-income families had better oral health owing to high calcium intake, whereas those from poor families had poor nutrition, which was associated with an increased risk of dental caries. Nutrient and income levels influence many diseases [22]. The WHO reported the mortality rates of non-infectious diseases by country, among which cardiovascular disease, cancer, and chronic respiratory diseases accounted for a large proportion [23]. Risk factors for cardiovascular diseases include hypertension, diabetes, obesity, and dyslipidemia. A study that included Vietnamese people found that low intake of vegetables and fruits and high sodium intake are risk factors for cardiovascular disease, as they increase the incidence of dyslipidemia [24]. In Italy, the Mediterranean diet pattern, which involves the consumption of an abundance of vegetables, olive oil, and fish, lowered the prevalence of metabolic syndrome as well as the risk of cardiovascular disease [25]. As a result, regardless of race and country, the nutrients consumed by income differ, which affects disease prevalence.

Similar to these studies, HL was also correlated with income level, and it was showed that nutritional intake according to income level influenced the onset of hearing loss. In the low-income group, a lower carbohydrate intake reduced the risk of HL (OR = 0.78, *p* = 0.031), as did a higher vitamin C intake (OR = 1.302, *p* = 0.0074). In the middle-income group, a lower protein intake reduced the risk of HL (OR = 0.638, *p* = 0.0442), while an increased fat intake reduced the risk of HL (OR = 1.332, *p* = 0.0432). A reduced vitamin B1 intake also decreased the risk of HL (OR = 0.746, *p* = 0.0401). In the high-income group, carbohydrate intake was a statistically significant variable; a low carbohydrate intake reduced the risk of HL (OR = 0.783, *p* = 0.0397). Therefore, income level should be considered when concerning the relationship between hearing impairment and nutrient intake. Studies reported that the incidence of HL is higher in low- and middle-income countries such as Cameroon, where nutrient intake is poor, compared to high-income countries. Furthermore, adolescents in underdeveloped countries were found to consume dairy products, meat, fruits, and vegetables as specified in the WHO dietary guidelines, but they suffer from vitamin A deficiency as they do not consume the required amount [26,27]. Low-income individuals are reportedly more vulnerable to noise exposure and environmental pollution. Air pollution is reportedly correlated with income inequality [28,29,30]. The known low is related to nutrient intake. As no study has investigated the association between nutrient intake and HL caused by income, this association should be analyzed as a risk factor for HL.

Our findings will be useful for the study of health policies related to HL. When establishing food- and nutrition-related policies or welfare programs, there is a need for differentiated nutrition interventions and management by social class. Active government support is also required, especially for low-income single-person households that are vulnerable to nutritional deficits. The need for food and nutrition support programs for the underprivileged has been recognized in many countries, with governments directly managing and operating such programs. In Korea, the Nutrition Plus Project was implemented in 2005 to establish a lifelong health management system for its citizens [31]. In the United States, the United States Department of Agriculture Rural Development runs the Supplemental Nutrition Assistance Program, Special Supplemental Nutrition Program for Women, Infants, and Children, Meals on Wheels, and Food Bank programs [32]. In Korea, individuals requiring assistance include pregnant women, lactating women, and infants, whereas, in the United States, they include children, older adults, and persons with disabilities. If introduced in Korea with reference to these policies, the rates of HL may be improved.

## 5. Conclusions

To prevent HL, different nutrients are important for each income group. Carbohydrate and vitamin C levels were higher in the low-income group; protein, fat, and vitamin B1 levels were crucial in the middle-income group; and carbohydrates were the main target nutrients for the high-income group. Overall, this study provides direction for HL-preventing public health policies.

## Figures and Tables

**Figure 1 nutrients-14-01655-f001:**
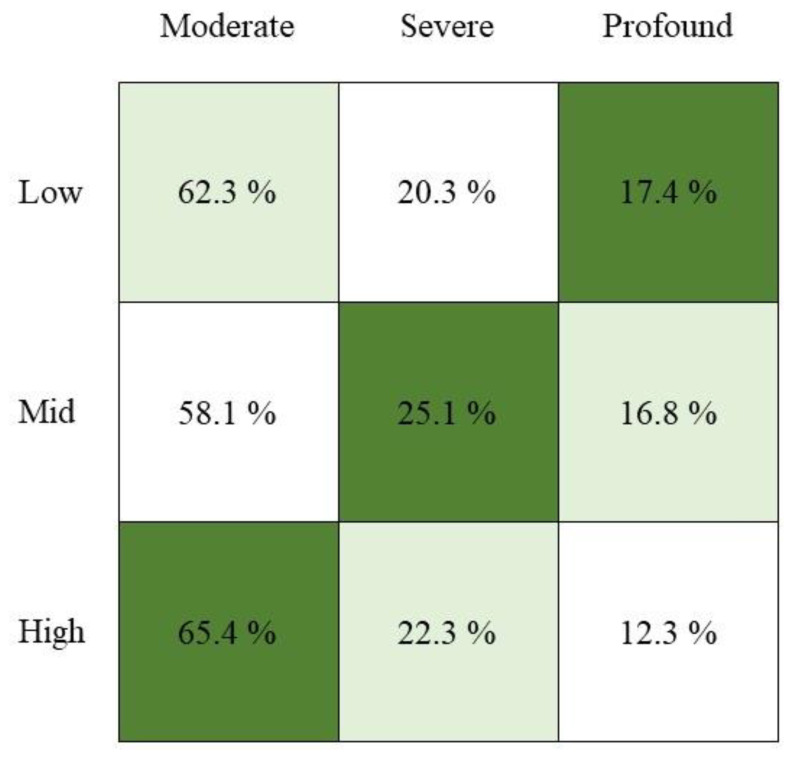
Degree of Hearing Loss in income group (*n* = 720).

**Table 1 nutrients-14-01655-t001:** Clinical characteristics of subjects (non-adjusted).

	No Hearing Impairment (*n* = 10,130)	Hearing Impairment (*n* = 720)	*p*-Value
Age			
20–24	611 (6.03%)	8 (1.11%)	**<0.0001**
25–29	774 (7.64%)	13 (1.81%)	
30–34	1057 (10.43%)	20 (2.78%)	
35–39	1495 (14.76%)	21 (2.92%)	
40–44	1298 (12.81%)	42 (5.83%)	
45–49	1112 (10.98%)	57 (7.92%)	
50–54	1309 (12.92%)	114 (15.83%)	
55–59	1181 (11.66%)	160 (22.22%)	
60–64	1293 (12.76%)	285 (39.58%)	
Sex (Male : Female)	3886:6244	357:363	**<0.0001**
Noise exposure (Yes : No)	1190:8940	125:595	**<0.0001**
Income			0.0508
Low	3964 (39.13%)	305 (42.36%)	
Middle	2041 (20.15%)	155 (21.53%)	
High	4125 (40.72%)	260 (36.11%)	
Marriage			**<0.0001**
Unmarried	7948 (78.46%)	611 (84.86%)	
Married	1071 (10.57%)	21 (2.92%)	
Divorced/Separated	1111 (10.97%)	88 (12.22%)	
Smoking			**<0.0001**
No smoker	6319 (62.38%)	383 (53.19%)	
Less than 5 packs	280 (2.76%)	11 (1.53%)	
Over 5 packs	3527 (34.82%)	325 (45.14%)	
Unknown	4 (0.04%)	1 (0.14%)	
Alcohol			**<0.0001**
No drink	2302 (22.72%)	213 (29.58%)	
2–4 times/month	5753 (56.79%)	335 (46.53%)	
2–4 times/week	1447 (14.28%)	99 (13.75%)	
Over 4 times/week	574 (5.67%)	69 (9.58%)	
Unknown	54 (0.53%)	4 (0.56%)	
Metabolic syndrome (Yes : No)	1867:8263	188:532	**<0.0001**
Waist (Yes : No)	3285:6845	275:445	**0.0015**
HDL-C (Yes : No)	3962:6168	294:426	0.3606
Fast glucose (Yes : No)	2301:7829	253:467	**<0.0001**
Hypertension (Yes : No)	1242:8888	142:578	**<0.0001**
Triglyceride (Yes : No)	2587:7543	233:487	**<0.0001**

Bold numbers highlight the statistical significance.

**Table 2 nutrients-14-01655-t002:** Multiple regression models including factors affects hearing impairment.

	B	OR (95% CI)	*p*-Value
Age	−0.443	0.642 (0.615–0.671)	**<0.0001**
Sex	0.2679	1.307 (1.025–1.667)	**0.0308**
Noise Exposure	−0.2501	0.779 (0.63–0.963)	**0.0208**
Income	0.0929	1.097 (1.004–1.199)	**0.0395**
Marriage	−0.1176	0.889 (0.788–1.002)	0.0549
Smoking	−0.0897	0.914 (0.81–1.031)	0.1444
Alcohol	0.0253	1.026 (0.941–1.118)	0.5665
Metabolic syndrome	0.0346	1.035 (0.864–1.24)	0.7076

Bold numbers highlight the statistical significance.

**Table 3 nutrients-14-01655-t003:** Statistical analysis results and standardized differences before and after propensity score matching.

Variables	Before Propensity Score Matching				After Propensity Score Matching			
	Observed Variables		*p*-Value	Standardized Difference	Observed Variables		*p*-Value	Standardized Difference
Covariates	Normal Hearing	Hearing Impairment			Normal Hearing	Hearing Impairment		
Age	44.08 ± 12.11	54.64 ± 9.60	<0.0001	−0.012963	54.47 ± 9.56	54.64 ± 9.60	0.9798	−0.026275
Sex	1.62 ± 0.49	1.5 ± 0.5	<0.0001	0	1.52 ± 0.5	1.5 ± 0.5	0.5467	0
Sound Exposure	0.12 ± 0.32	0.17 ± 0.38	<0.0001	0	0.17 ± 0.38	0.17 ± 0.38	1.00	0

**Table 4 nutrients-14-01655-t004:** Clinical characteristics of subjects (1:3 PS matching).

	No Hearing Impairment (*n* = 2160)	Hearing Impairment (*n* = 720)	*p*-Value
Age	54.47 ± 9.56	54.64 ± 9.60	
20–24	24 (1.11%)	8 (1.11%)	0.9798
25–29	39 (1.81%)	13 (1.81%)	
30–34	60 (2.78%)	20 (2.78%)	
35–39	63 (2.92%)	21 (2.92%)	
40–44	126 (5.83%)	42 (5.83%)	
45–49	171 (7.92%)	57 (7.92%)	
50–54	342 (15.83%)	114 (15.83%)	
55–59	480 (22.22%)	160 (22.22%)	
60–64	855 (39.58%)	285 (39.58%)	
Sex (Male : Female)	1043:1117	357:363	0.5467
Noise exposure (Yes : No)	375:1785	125:595	1.00
Income			**0.0012**
Low	789 (36.53%)	305 (42.36%)	
Middle	425 (19.68%)	155 (21.53%)	
High	946 (43.8%)	260 (36.11%)	
Marriage			0.2692
Unmarried	1883 (87.18%)	611 (84.86%)	
Married	58 (2.69%)	21 (2.92%)	
Divorced/Separated	219 (10.14%)	88 (12.22%)	
Smoking			**0.0444**
No smoker	1215 (56.25%)	383 (53.19%)	
Less than 5 packs	53 (2.45%)	11 (1.53%)	
Over 5 packs	892 (41.3%)	325 (45.14%)	
Unknown	0 (0%)	1 (0.14%)	
Alcohol			0.4389
No drink	584 (27.04%)	213 (29.58%)	
2–4 times/month	1053 (48.75%)	335 (46.53%)	
2–4 times/week	332 (15.37%)	99 (13.75%)	
Over 4 times/week	180 (8.33%)	69 (9.58%)	
Unknown	11 (0.51%)	4 (0.56%)	
Metabolic syndrome (Yes : No)	544:1616	188:532	0.6212
Waist (Yes : No)	847:1313	275:445	0.6274
HDL-C (Yes : No)	874:1286	294:426	0.8609
Fast glucose (Yes : No)	734:1426	253:467	0.5709
Hypertension (Yes : No)	402:1758	142:578	0.5095
Triglyceride (Yes : No)	690:1470	233:487	0.8356

Bold numbers highlight the statistical significance.

**Table 5 nutrients-14-01655-t005:** Multiple regression models including factors affects hearing impairment.

	B	OR (95% CI)	*p*-Value
Age	0.00219	1.002 (0.957–1.05)	0.9263
Sex	0.1226	1.13 (0.857–1.491)	0.3847
Noise Exposure	−0.0493	0.952 (0.758–1.195)	0.6706
Income	−0.1571	0.855 (0.776–0.941)	**0.0013**
Marriage	0.0822	1.086 (0.951–1.24)	0.2251
Smoking	0.1414	1.152 (1.004–1.322)	**0.0439**
Alcohol	−0.037	0.964 (0.879–1.057)	0.4308
Metabolic syndrome	0.0325	1.033 (0.85–1.256)	0.7442

Bold numbers highlight the statistical significance.

**Table 6 nutrients-14-01655-t006:** Differences in nutritional intake between normal hearing and hearing loss according to amount of income (Simple regression).

Income	Low			Middle			High		
	No Hearing Impairment (*n* = 789)	Hearing Impairment (*n* = 305)	*p*-Value	No Hearing Impairment (*n* = 425)	Hearing Impairment (*n* = 155)	*p*-Value	No Hearing Impairment (*n* = 946)	Hearing Impairment (*n* = 260)	*p*-Value
Energy	198,167 ± 79,111	198,888 ± 81,235	0.8373	206,583 ± 80,248	205,286 ± 79,328	0.9376	209,248 ± 86,065	218,856 ± 82,982	0.1592
Water	94,686 ± 59,402	90,417 ± 61,848	0.2831	104,538 ± 58,422	97,997 ± 62,154	0.0794	116,389 ± 67,915	119,229 ± 75,367	0.5981
Protein	69.91 ± 36.16	70.23 ± 48.85	0.2474	74.46 ± 36.03	70.4 ± 29.5	0.4068	77.46 ± 39.6	77.15 ± 34.14	0.7115
Fat	36.29 ± 32.51	34.12 ± 28.63	0.5555	40.87 ± 30.83	35.69 ± 26.49	0.2164	41.77 ± 30.96	40.62 ± 26.51	0.9910
Carbonhydrates	32,842 ± 12,332	33,673 ± 12,219	0.6251	33,337 ± 11,906	34,012 ± 11,677	0.6679	334 ± 126.63	36,086 ± 12,974	**0.0023**
Fiber	7.83 ± 5.12	7.65 ± 4.57	0.5205	8.53 ± 5.48	8.08 ± 4.6	0.3712	9.15 ± 7.28	9.57 ± 6.54	0.3308
Ash	20.04 ± 10.93	19.53 ± 10.69	0.3560	22.08 ± 10.92	20.9 ± 9.36	0.7155	21.97 ± 11.73	23.39 ± 12.21	0.0678
Calcium	50,686 ± 31,697	50,376 ± 36,478	**0.0196**	55,164 ± 318.3	53,821 ± 29,757	0.5612	55,289 ± 311.16	58,586 ± 29,999	0.1092
Phosphate	11,899 ± 50,276	119,259 ± 60,298	0.9027	125,656 ± 51,515	119,731 ± 41,279	0.4406	12,893 ± 54,923	131,296 ± 49,319	0.5153
Iron	16.29 ± 16.2	15.49 ± 11.69	0.5296	16.63 ± 9.93	15.22 ± 8.49	0.1420	17.27 ± 11.53	18.35 ± 11.76	0.0837
Sodium	490,983 ± 321,362	489,789 ± 319,621	0.3766	526,527 ± 311,536	533,635 ± 290,796	0.4896	505,544 ± 307,025	547,122 ± 362,251	0.1177
Kalium	309,333 ± 150,141	301,197 ± 169,215	0.3223	338,218 ± 151,778	313,654 ± 135,12	0.1995	354,18 ± 175,317	366,578 ± 168,635	0.0585
Vitamin A	83,185 ± 88,315	78,983 ± 91,335	0.1560	88,613 ± 78,632	84,512 ± 98,073	0.1501	96,595 ± 104,318	104,737 ± 117,023	0.3533
Carotin	435,043 ± 468,906	392,383 ± 391,069	0.2117	466,841 ± 446,398	411,566 ± 443,996	0.2188	506,955 ± 573,327	550,608 ± 674,964	0.5929
Retinol	11,721 ± 62,145	89.13 ± 24,183	0.1252	87.95 ± 11,458	82.42 ± 91.4	0.6173	11,286 ± 372.4	10,683 ± 27,257	0.5565
Vitamin B1	1.34 ± 0.84	1.34 ± 0.96	0.5003	1.37 ± 0.71	1.3 ± 0.56	0.4990	1.43 ± 0.74	1.46 ± 0.73	0.2186
Riboflavin	1.2 ± 0.78	1.17 ± 0.8	0.5354	1.29 ± 0.69	1.19 ± 0.57	**0.0184**	1.32 ± 0.68	1.37 ± 0.66	0.3483
Niacin	16.78 ± 9.05	16.71 ± 10.85	0.7218	18.1 ± 9.55	16.29 ± 6.82	0.1372	18.69 ± 9.85	19.02 ± 9.15	0.6980
Vitamin C	11,231 ± 88.02	10,342 ± 97.2	**0.0103**	11,953 ± 94.48	11,209 ± 79.82	0.3674	13,424 ± 10,724	13,839 ± 10,052	0.5331

Bold numbers highlight the statistical significance.

**Table 7 nutrients-14-01655-t007:** Differences in nutritional intake between normal hearing and hearing loss according to amount of income (multiple regression).

Income	Low			Middle			High		
	B	OR (95%CI)	*p*-Value	B	OR (95%CI)	*p*-Value	B	OR (95%CI)	*p*-Value
Energy	0.1924	1.212 (0.897–1.637)	0.2099	0.0327	1.033 (0.679–1.572)	0.8789	−0.0656	0.936 (0.689–1.272)	0.6745
Water	−0.1141	0.892 (0.743–1.072)	0.2233	0.1048	1.11 (0.839–1.47)	0.4638	−0.0245	0.976 (0.799–1.191)	0.8102
Protein	0.0828	1.086 (0.801–1.474)	0.5946	−0.4495	0.638 (0.412–0.988)	**0.0442**	−0.0353	0.965 (0.713–1.307)	0.8195
Fat	−0.0757	0.927 (0.757–1.135)	0.4634	0.2867	1.332 (1.009–1.759)	**0.0432**	0.1332	1.142(0.935–1.396)	0.1932
Carbonhydrates	−0.2483	0.78 (0.623-0.978)	**0.031**	−0.2884	0.749 (0.54–1.04)	0.0846	−0.2448	0.783 (0.62–0.989)	**0.0397**
Fiber	−0.0833	0.92 (0.765–1.106)	0.3754	−0.1794	0.836 (0.633–1.104)	0.2067	0.0432	1.044 (0.843–1.293)	0.6926
Ash	−0.0687	0.934 (0.7–1.245)	0.6405	0.2245	1.252 (0.858–1.825)	0.2434	−0.196	0.822 (0.608–1.111)	0.2019
Calcium	0.0133	1.013 (0.836–1.229)	0.8921	−0.1142	0.892 (0.674–1.181)	0.4245	−0.0806	0.923 (0.75–1.135)	0.4453
Phosphate	−0.0225	0.978 (0.717–1.334)	0.8872	0.2804	1.324 (0.833–2.102)	0.235	0.1413	1.152 (0.831–1.597)	0.3972
Iron	0.0988	1.104 (0.902–1.352)	0.3384	0.1496	1.161 (0.873–1.545)	0.3044	−0.0681	0.934 (0.754–1.157)	0.5331
Sodium	−0.0277	0.973 (0.782-1.211)	0.8041	−0.2653	0.767 (0.577–1.02)	0.0682	0.0985	1.104 (0.889–1.371)	0.3729
Kalium	0.1047	1.11 (0.845–1.459)	0.452	0.2169	1.242 (0.868–1.777)	0.2352	0.0911	1.095 (0.813–1.475)	0.5488
Vitamin A	0.0833	1.087(0.869–1.359)	0.4649	0.0163	1.016 (0.746–1.384)	0.9175	−0.0635	0.938 (0.747–1.179)	0.5855
Carotin	−0.0325	0.968 (0.741–1.265)	0.8122	0.0855	1.089 (0.746–1.591)	0.6583	0.1156	1.123 (0.864–1.458)	0.3865
Retinol	0.1176	1.125 (0.962–1.315)	0.1395	−0.1897	0.827 (0.66–1.036)	0.0992	0.1245	1.133 (0.963–1.333)	0.1335
Vitamin B1	0.0351	1.036 (0.847–1.266)	0.7319	−0.2927	0.746 (0.564–0.987)	**0.0401**	−0.0413	0.96 (0.767–1.2)	0.7174
Riboflavin	−0.1877	0.829 (0.653–1.053)	0.1238	0.0967	1.101 (0.782–1.551)	0.58	−0.1872	0.829 (0.652–1.055)	0.1271
Niacin	−0.0164	0.984 (0.776–1.247)	0.8922	0.254	1.289 (0.906–1.834)	0.1576	0.1341	1.143 (0.9–1.453)	0.2727
Vitamin C	0.2637	1.302 (1.073–1.579)	**0.0074**	−0.00033	1 (0.771–1.296)	0.998	0.0685	1.071 (0.883–1.298)	0.4855

Bold numbers highlight the statistical significance.

## Data Availability

The data used in this study are available for free on the KNHANES website (Available online: https://knhanes.kdca.go.kr/knhanes/sub03/sub03_02_05.do, accessed on 15 September 2021) for academic research.

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
