# Peer review of "Relationship between Nutrient Intake and Hearing Loss According to the Income Level of Working-Aged Adults: A Korean National Health and Nutrition Survey"

_nutrients, 2022, doi:10.3390/nu14081655_

Round 1
Reviewer 1 Report
The authors present an hughe analyse of intake of any minerals and vitamins regarding to hearing loss.
Hearing loss present a large reason of that in each population. Mostly associated to the noise,, genetic bases (pendred syndroma, EVA syndroma etc.), and any toxins in water and food.
The intake of vitamins according the antioxidants associate to general helath status and oxidative stress especially.
We noe that tha higher intake of fat associate to "vessel disease or endotel pathology" . The lot of that depend on intake of omega3 unsaturated fat acids asspecially and overintake the saturated fat acids like the palm oil, coconuts oil, peanuts oil and other oils used in coussine of the world.
On the second hand thy low intake of vitamin D has an "epidemic" charkter worldwide and tha last analyses dhowed that the recommand intake for blood serum 75 mikromol/l is low for health status on the world.
I thanks to the authors for the large analyse, but i must recommand the large inrodution to reason of the hearing loss. I hope that the nutrition has not the large data for the evidence of the reason of HL and low nutrition facts.
On the second hand the data about the vascular risk for the HL like as the vessel disease stroke, heart attack and etc., are describedn in literature ( RED MID study in US).
I am not sure that the consludions are supported by results of analyses so strongly.
Reviewer 2 Report
Line 16: why South Korea is mentioned and why not also in lines 7-15 instead of Korea, also in line 49.
Line 24: After propensity score matching: insufficient English, please correct!
Line 36: nutrient intake by income bracket: unclear English, please adjust!
Line 33: in this stage the relation between nutrition and hearing impairment apparently remains uncertain, it should be good to explain this uncertainty (if possible) or to make it less uncertain.
Line 57: what is meant by “food insecurity”, please add short explanation, same in line 82 for “safety”.
Line 65: food “of good quality” should be mentioned.
Line 67: it is not clear what is meant with “There was a sense of safety”.
Line 68: in “Materials and Methods” nothing is mentioned about the hearing impairment data, why? How did participants know & report their hearing status upon which measures, please clarify!
Line 103: “classified it into five to three categories” is a little puzzled, please, rectify!
Line 108: 60-64 years is the last group, why not groups with elderly people until e.g. until 99 years?
Line 122: in this paragraph “metabolic syndrome” it is not clear how participants know their data?
Line 134: What is the reliability of “food intake frequency survey”, how did participants know data?
Please try to explain!
Line 188: “HL was more likely to occur in women than men” which is not common, the opposite is!
Please try to explain or bring into discussion!
Line 194: “heating” must be “hearing”.
Line 217: “metallic syndrome” must be “metabolic syndrome”.
Line 227: “HL”, please, don’t use an abbreviation for this first appearance, “HL” in brackets!
Line 243: “OR”, the same remark as for HL in line 227.
Line 262: In the discussion paragraph a lot of data are presented but almost nothing is in discussion. One should expect not only statements but also a lot of explanations, what is the expected reason for the gathered results and comparisons with literature, or what is not, please discuss!
